# Inheritances, social classes, and wealth distribution

**Pedro Patrício**[1,2]*, **Nuno A. M. Araújo**[2,3]

**1** Departamento de Física, ISEL - Instituto Superior de Engenharia de Lisboa, Instituto Politécnico de Lisboa, Lisboa, Portugal, **2** Centro de Física Teórica e Computacional, Faculdade de Ciências, Universidade de Lisboa, Lisboa, Portugal, **3** Departamento de Física, Faculdade de Ciências, Universidade de Lisboa, Lisboa, Portugal

* pedro.patricio@isel.pt

## Abstract

We consider a simple theoretical model to investigate the impact of inheritances on the wealth distribution. Wealth is described as a finite resource, which remains constant over different generations and is divided equally among offspring. All other sources of wealth are neglected. We consider different societies characterized by a different offspring probability distribution. We find that, if the population remains constant, the society reaches a stationary wealth distribution. We show that inequality emerges every time the number of children per family is not always the same. For realistic offspring distributions from developed countries, the model predicts a Gini coefficient of $G \approx 0.3$. If we divide the society into wealth classes and set the probability of getting married to depend on the distance between classes, the stationary wealth distribution crosses over from an exponential to a power-law regime as the number of wealth classes and the level of class distinction increase.

## I. Introduction

Empirical wealth distributions are characterized by two enduring features. For the large majority of the population, which has small or medium wealth $w$, the distribution is positively skewed, roughly resembling a lognormal distribution. However, the tail for the wealthier is well approximated by a power-law distribution [1]:

$$f(w) \propto \frac{1}{w^\alpha}, \tag{1}$$

also referred to as a Pareto law. Although this law refers only to the wealthier and, therefore, to a small percentage of the population, its importance may not be overlooked, as it concerns the richest part of the population, holding the larger percentage of the total wealth. The more unequal the society is, the smaller is the value of $\alpha$. The data regarding labor income is now very well documented, and the corresponding $\alpha$ varies between 1.5 and 3 [2, 3]. The past forty years have seen a disturbing increase in income inequality (and consequently smaller values of $\alpha$) almost everywhere in the world [4]. General wealth distributions are difficult to find, as they concern all material assets, in the form of real property and financial claims. Nevertheless,

**Data Availability Statement:** All relevant data are within the manuscript.

**Funding:** We acknowledge financial support from the Portuguese Foundation for Science and Technology (FCT) under Contracts no. UIDB/00618/2020 and UIDP/00618/2020. The funders

had no role in study design, data collection and analysis, decision to publish, or preparation of the manuscript.

**Competing interests:** The authors have declared that no competing interests exist.

almost all studies find that the wealth distribution is more unequal than the labor income distribution [5].

The ubiquitous Pareto law, which also appears in other socio-economic contexts, such as firm or city sizes [6, 7], hints at some universality, which should be robust to the fine details of the theoretical model we use to describe a society. Many models have been proposed to explain the tail distribution of wealth [8–11] (or, more recently [12, 13], in the context of physics), mainly along the lines of proportional random growth, which assumes Gibrat's law of proportionate effect. This law states that the distribution of the percentage growth rate of a unit (e.g. wealth, the size of a firm or a city) is independent of its size.

If one aims at an understanding of the forces that contribute to larger or smaller wealth inequalities, the explicit mechanisms behind wealth inequality must be incorporated. This rapidly leads to complex models that are difficult to analyze. Indeed, the reasons behind wealth inequality are innumerable: we have, first of all, the inheritance and education we receive from our parents, the marriages or alliances we make, associated so many times with the relatively closed circles of relationships we establish, our business talent and ability to work, our age and health or simply mere luck. This article does not intend to make an extensive literature review about these economic models. The interested reader is referred to Refs. [5, 14], for a more economical perspective and Ref. [15], for a more physical one.

For the sake of simplification, these models may be divided into two types. Lifecycle models (LCM) consider the wealth evolution during an individual lifetime, in which inheritances play no role. These are also known as intragenerational models. Other models suppress interest in lifecycle variations and focus on intergenerational links. Very few contributions have attempted to deal simultaneously with both the lifecycle and inherited components of wealth [5].

The simple model proposed in our article in the context of statistical physics belongs to the second type: we intend to quantify the evolution of the distribution of wealth over several successive generations. For each cohort, the sum of the wealth of all individuals is considered to be constant. In our model, wealth could be thought as a finite resource of the society, which remains constant and must be divided by all individuals. Of all possibilities for enrichment, we will focus on two particular main aspects. On the one hand, the variable number of children of each family, which implies different inheritances. If we assume that the inheritance is equally divided by all children, the smaller the number of children, the greater the inheritance of each child. On the other hand, the fact that people tend to marry people with comparable wealth, or belonging to the same social circle or class. These two factors will inevitably lead to an unequal distribution of wealth, even if we start from a very homogeneous society.

One of the first intergenerational models [16] is closely related to the model we present here. It considers a simplified society in which every family has exactly two children, a boy and a girl. This model discusses the implications for wealth inequality of primogeniture, when all the family fortune is given to the male heir, equal division, or unequal division. It also discusses the effect of having random mating, in which there is no relation between the wealth of the husband and the wife, class mating, in which the wife has exactly the same wealth of the husband, or an assortative mating, something in between. However, this model never discusses other offspring distributions, as we do in our article. Other intergenerational models considered societies with individuals of different age, with a mortality probability distribution [17], or other complex features regarding personal earnings, consumption, savings and motives of bequests [14, 18].

This paper is organized as follows. In section II, we present our intergenerational model, which is characterized by a particular marriage and offspring probability distributions. In section III, we describe the results we obtain for a society without and with well defined classes.

Particular emphasis is given to the stationary wealth distributions we determine in each case. In the last section, we discuss the importance of our results both in the context of economic inequality and statistical physics.

## II. Model

We consider a society composed of $N_0$ individuals with an initial distribution of wealth and gender. For individual $i$ ($i = 1, \ldots, N_0$), the wealth $w_i$ is drawn from an initial *wealth* probability distribution $f_0^w(w)$. The gender $g_i$ is either "female" or "male", with equal probability.

We consider that marriages among people with comparable wealth are more likely. Thus, we organize the society into $N_c$ classes with $N_0/N_c$ individuals each. Individuals are organized into classes following the rank of increasing wealth. For simplicity, we only consider different-gender marriages. If $i$ and $j$ are of a different gender, the probability of getting married $f_{ij}^m$ is

$$f_{ij}^m \propto e^{-\beta d_{ij}}, \qquad (2)$$

where $d_{ij}$ is defined as a distance between their classes and $\beta$ is the level of class distinction, or the inverse of the level of mixing between classes. For $\beta = 0$, $f_{ij}^m$ is the same for all pairs and there is no distinction between classes. The larger the value of $\beta$ is, the more likely it is that marriage between individuals in the same class are favored over inter-class marriage. Here, we consider $d_{ij} = |c_j - c_i|$, where $c_i$ and $c_j$ is the rank of the class, when they are all ordered by increasing wealth.

We select the pairs to couple in the following way. For each individual $i$, we randomly select from which class the couple $j$ is, where the probability $p(c_j)$ for each class $c_j$ is,

$$p(c_j) = \frac{n(c_j) f_{ij}^m}{\sum_{c_k} n(c_k) f_{ij}^m}. \qquad (3)$$

$n(c_j)$ is the number of individuals in class $c_j$ that are of a different gender than $i$ and the sum is over all classes. We then randomly select one individual $j$ to marry $i$ among the $n(c_j)$. Note that, instead of using classes, we could think of a marriage probability distribution that depends directly on the wealth difference $d_{ij} = |w_j - w_i|$. However, this methodology, which is equivalent in the limit $N_c \rightarrow N_0$, is computationally much more demanding.

Once all couples are defined, a new generation of individuals is generated. Each married couple $ij$ is replaced by $o_{ij}$ offspring according to a specific *offspring* probability distribution $f^o(o)$, and the total wealth of the parents is equally distributed among the offspring. Each individual of the new generation is either a "female" or a "male" with equal probability. Here, we represent the complete offspring discrete value distribution by the set

$$f^o = \{f^o(0), f^o(1), \ldots, f^o(n_{\max})\}, \qquad (4)$$

where $n_{\max}$ is the maximum number of offspring per family.

Due to the stochastic nature of the dynamics, for each generation, the number of "female" and "male" individuals is only equal on average. Thus, at the end of the matching protocol some excess individuals of a given gender will be unpaired or without offspring. We redistribute their wealth equally among all individuals of the new generation. So, the total wealth is conserved at all times.

Once a new society with $N_1$ individuals is formed, classes are redefined according to the new wealth distribution $f_1^w(w)$, and the process of generating the next generation is repeated.

## III. Results

### A. Societies without classes

Let us first consider a society of $N_0 = 10^5$ without classes, where all pairs of individuals of a different gender are equally likely to get married, i.e., $N_c = 1$ or $\beta = 0$. We set

$$f^o = \{0, 0, 1\} \quad , \tag{5}$$

which corresponds to exactly two offspring per couple and the size of the society remains approximately constant. To characterize the level of wealth inequality, we compute the Gini coefficient $G$, defined as,

$$G = 1 - \frac{2}{N^2} \sum_{i=1}^{N} \sum_{j=1}^{i} \frac{w_j}{\mu} \tag{6}$$

where $N$ is the size of the population, $\mu$ the average wealth and the sum follows the rank of increasing wealth. $G = 0$ for an egalitarian society, where $w_i$ is the same for all individuals, and $G \approx 1$ for a large society where all the wealth is concentrated in a few number of individuals.

We set the initial distribution of wealth to be uniform, of average $\mu$, with $f_0^w(w_i) = 1/(2\mu)$ and $0 < w_i < 2\mu$, which corresponds to $G = 1/3$. Fig 1 shows the wealth distribution for four different generations. The Gini coefficient rapidly converges to zero. This is in fact the case for any initial wealth distribution. Since individuals are paired at random and their wealth evenly distributed among their two offspring, at each iteration pairwise heterogeneities in the wealth distribution are reduced. Precisely, the average wealth remains $\mu$ at all times, but the variance in generation $g$ is $\sigma_g^2 = \sigma_0^2/2^g$, which vanishes asymptotically. So, for any initial distribution of wealth, the society rapidly converges towards an egalitarian society where all individuals have (approximately) the same wealth. This result was also obtained by Blinder [16], in his intergenerational model in which each family had two children, a boy and a girl.

In fact, equality emerges for any society in which each couple has exactly the same number of children $n$. If $n < 2$ ($n > 2$), the population decreases (increases) exponentially. However, the result $(\sigma_g/\mu_g)^2 = (\sigma_0/\mu_0)^2/2^g$ still holds.

We now consider a society with the same $N_0$ but a different offspring probability distribution,

$$f^o = \{0, 1/3, 1/3, 1/3\} . \tag{7}$$

Note that, since the average offspring per couple is two, the size of the society remains (approximately) constant. Fig 2 shows the wealth distribution for different generations. Starting from a Gaussian distribution of average $\mu$ and $\sigma = \mu/10$, which yields a Gini coefficient $G \approx 0.05$. In the first generation, the distribution is characterized by a sequence of three peaks, which correspond to individuals from families with one, two, and three offspring. The third peak (higher wealth) has average $2\mu$ and includes $1/6$ of all individuals, which are the ones from families with only one offspring. The peak in the middle, is centered at $\mu$ and accounts for $1/3$ of the population, corresponding to the individuals from families with two offspring. The peak on the left has average $\mu/3$ and the narrowest dispersion and corresponds to the $1/2$ of the population, belonging to families with three offspring. After a few generations, the wealth distribution rapidly converges to a well-defined distribution, as shown for generation ten in the figure. The Gini coefficient increases with the generation and converges to $G \approx 0.35$ after a few iterations. This suggests that wealth inequality is observed even for a society without classes, provided that the number of offspring per family is not always two. Fig 3 shows the fraction of wealth distributed between three different groups: the 10% richer, the middle 40%, and the 50%

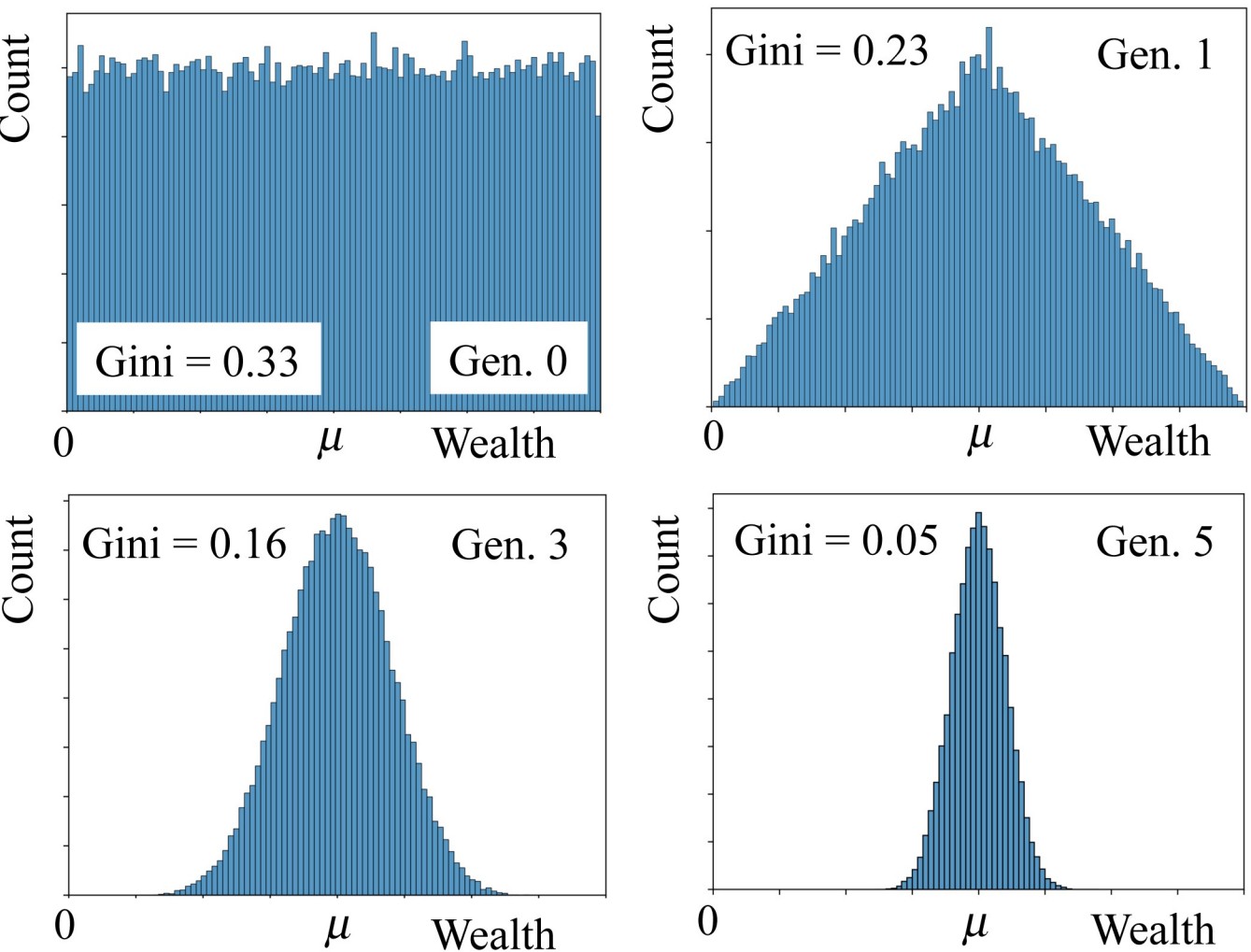

**Fig 1. Evolution of the wealth distribution for a society without classes and exactly two offspring per couple.** Initially, the wealth distribution is uniform, $f_0^w(w_i) = 1/(2\mu)$, for $0 < w < 2\mu$ (left-upper panel). In the first generation the wealth distribution is triangular (right-upper panel). For later generations $g$, the wealth distribution converges to a Gaussian distribution, with average $\mu$ and variance $\sigma_g^2 = \sigma_0^2/2^g$, where $\sigma_0$ is the initial variance. Thus, the Gini coefficient vanishes exponentially.

poorer. The first group has 27% of the total wealth, which is slightly more than the third group, which means that, on average, an individual of the first group have five times more wealth than one from the third group.

The shape of the wealth distribution reported for generation ten is in fact very robust and corresponds to a stationary distribution. We simulated higher generations and found no visual differences between the distributions. We have also considered other initial configurations, such as, for example, uniform and a bimodal distribution and, for all of them we obtained the same stationary wealth distribution, after a proper rescaling by the average wealth $\mu$. Fig 4 shows this stationary wealth distribution in a linear-linear and a log-linear scale. It is clear that, for large values of the wealth, the distribution decays logarithmic. For the sake of comparison, we also represented in a solid line a log-normal distribution with the same average wealth and variance. The main features of the wealth distribution are well captured by a log-normal distribution. Notwithstanding, the log-normal has a larger population for lower values and a slightly lower peak.

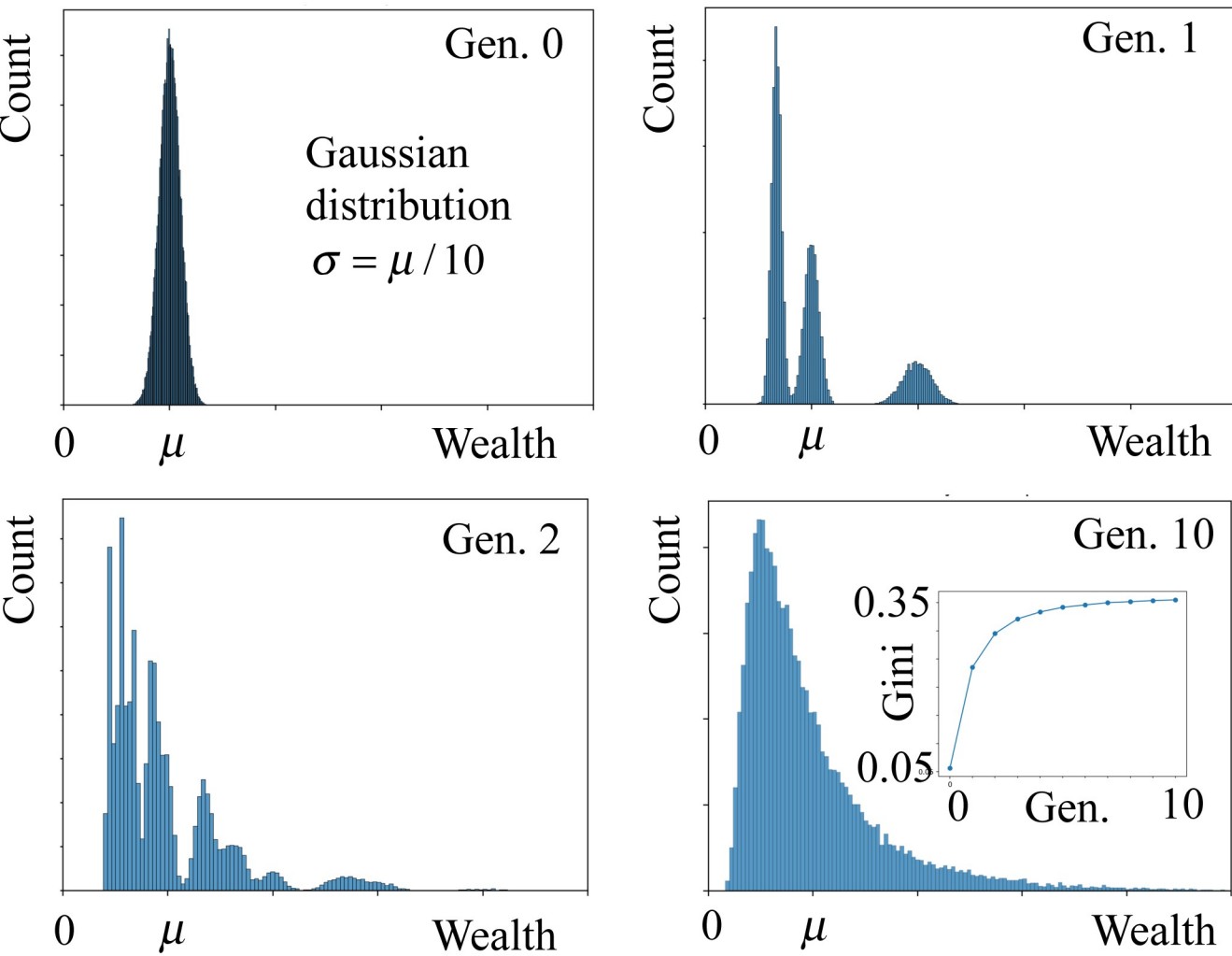

**Fig 2. Evolution of the wealth distribution for a society without classes and $f^c$ = {0, 1/3, 1/3, 1/3}.** Initially, the wealth distribution is a Gaussian of average $\mu$ and $\sigma = \mu/10$ (left-upper panel). In the first generation (right-upper panel), the distribution is a sequence of three peaks, which correspond to individuals from families with one, two, and three offspring. In the second generation (left-bottom panel), each peak is split into three. The wealth distribution converges rapidly to a broad stationary distribution, as shown in the right-bottom panel for generation ten. In the inset is the evolution of the Gini coefficient with the generation, which in three generations goes from $\approx 0.05$ to $\approx 0.35$.

We investigated several "societies" with other offspring probability distributions, including more realistic distributions, such as:

$$f^o = \{0.1, 0.2, 0.4, 0.2, 0.1\} \quad . \tag{8}$$

For this particular choice, which follows approximately the recent statistical bulletin [19], but in which the size of the society remains constant, the Gini coefficient rapidly converges to $G \approx 0.28$. We also considered some eccentric offspring distributions. If the size of the society remains constant, these lead to stationary wealth distributions with smaller or larger Gini coefficients. As a general rule, as the variety of the number of offspring per family increases, also increases the wealth inequality.

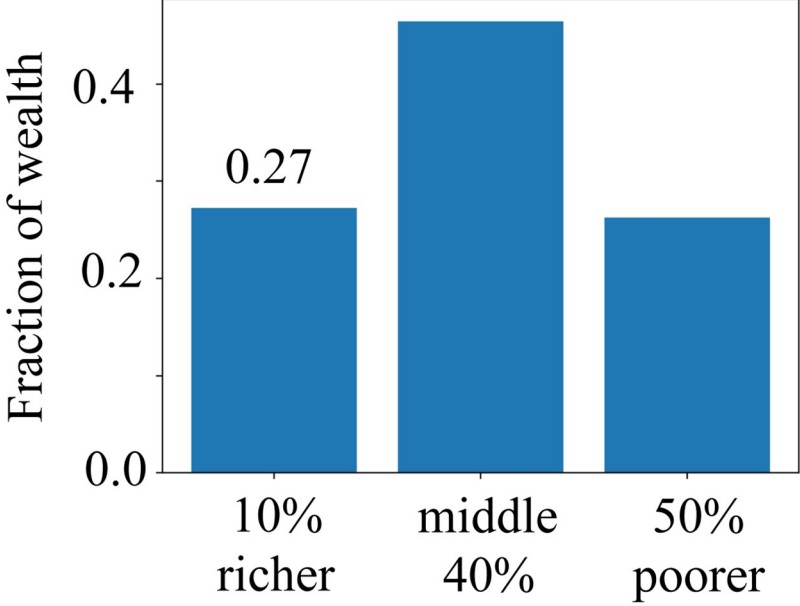

**Fig 3. Distribution of the wealth among three different groups: 10% richer, the middle 40%, and the 50% poorer.**
Results are for generation ten in the society of Fig 2, which are a good approximation of the stationary distribution.

## B. Societies with classes

We now study the impact of having a probability of getting married that depends on the class of each individual (see section II). For simplicity, we consider the representative offspring probability distribution where each couple has equal probability to have one, two, or three offspring, which corresponds to $f^o = \{0, 1/3, 1/3, 1/3\}$.

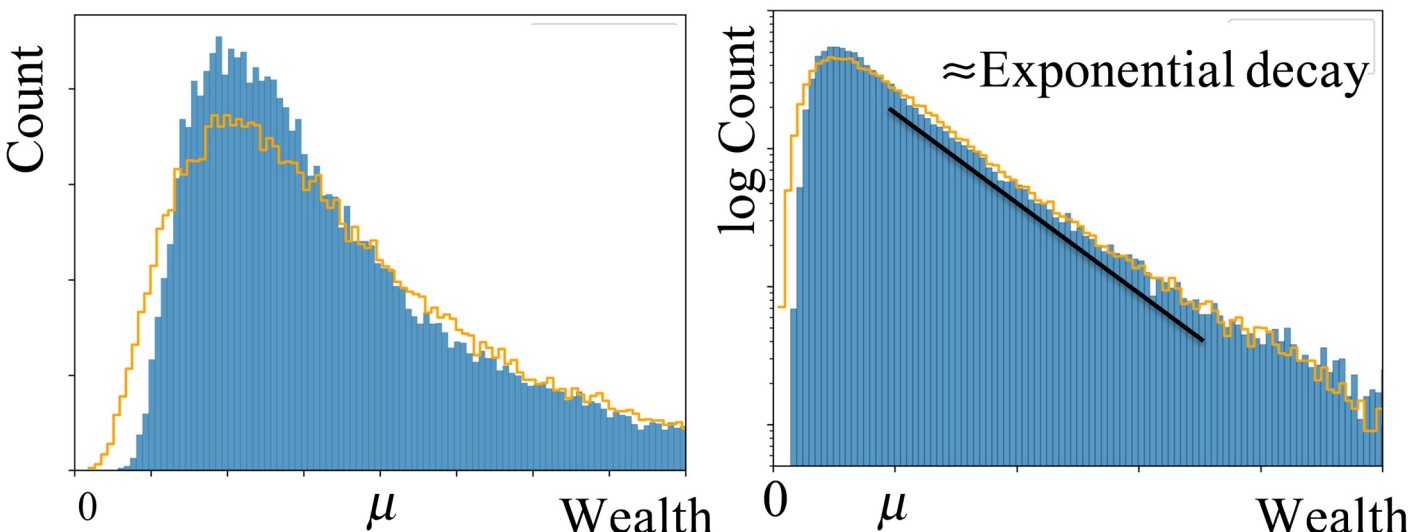

**Fig 4. Stationary wealth distribution for a society without classes, in a linear-linear (left) and a log-linear (right) scale.** Results are for generation ten in the society of Fig 2. The (orange) solid line corresponds to a log-normal distribution with the same average wealth and variance. In the right panel, the (black) solid line corresponds to an exponential decay with a characteristic wealth of $w_c \approx 1, 6\mu$.

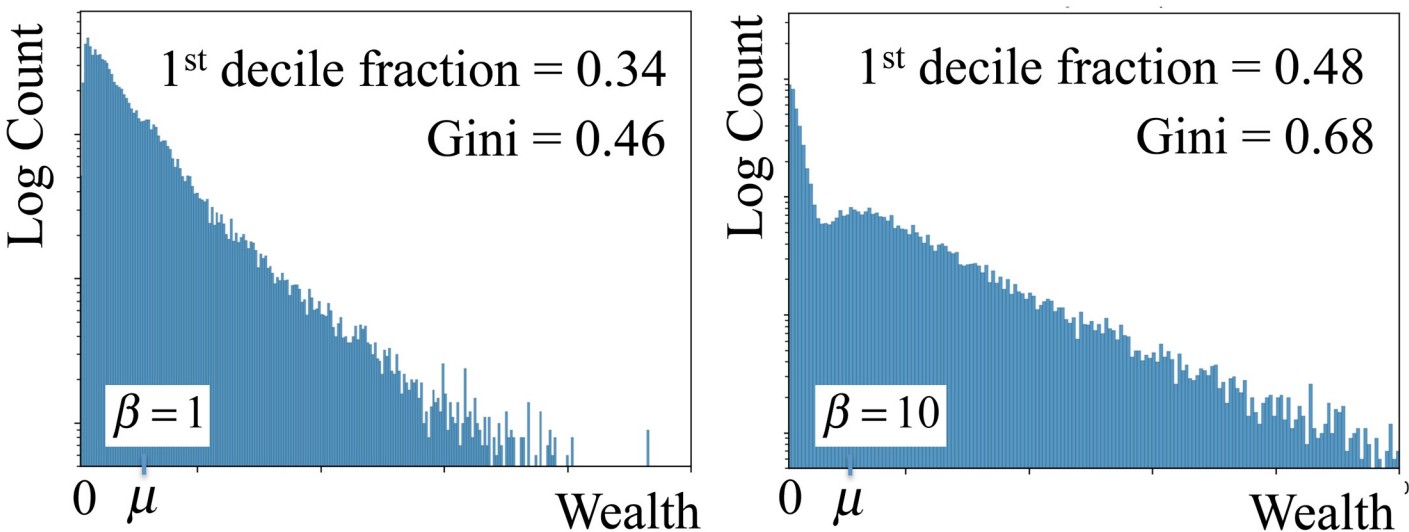

**Fig 5. Stationary wealth distributions for a society with three classes for $\beta = 1$ (left) and $\beta = 10$ (right).** For $\beta = 1$, the Gini coefficient is 0.46 and the 10% richest individuals accumulate 34% of the total wealth. For $\beta = 10$, the Gini coefficient is 0.48 and the top 10% accumulate 48% of the total wealth.

Let us consider first a society with $N_c = 3$ classes, where the poorer $N_0/3$ individuals are in the lower class, the next $N_0/3$ in the middle one, and the richer $N_0/3$ individuals in the upper one. As explained in section II, we consider a marriage probability between individuals $i$ and $j$ proportional to $\exp(-\beta d_{ij})$, where $d_{ij} = |c_j - c_i|$ is the difference between the number of their classes. For $\beta = 1$, for a society with the same number of individuals per class, the probability for an individual to marry someone from the same class is only $e \approx 2.7$ larger than the one of marrying with someone from a neighboring class. However, for $\beta = 10$, this factor increases by four orders of magnitude and so it is practically impossible to have marriages between individuals of different classes, except when someone in a class is left without a pair.

We start with $N_0 = 2 \times 10^5$ and a Gaussian wealth distribution of average $\mu$ and $\sigma = \mu/10$. As before, the wealth distribution rapidly converges after a few generations. Fig 5 shows the stationary wealth distribution for $\beta = 1$ (obtained at generation 10) and $\beta = 10$ (obtained at generation 20). The distribution for the society with higher degree of mixing between classes ($\beta = 1$) is similar to the one found for a society without classes in Fig 3, but with a higher Gini coefficient ($G \approx 0.46$) and with 34% of the wealth concentrated in the top 10% of the population. For the society with lower degree of mixing ($\beta = 10$) the inequalities are even more evident. The Gini coefficient is $G \approx 0.68$ and the top 10% accumulate 48% of the total wealth.

Fig 6 shows the stationary wealth distribution for different values of $\beta$. For $\beta = 0$, the wealth distribution shows a well-defined peak around the average wealth $\mu$, as discussed before. As $\beta$ increases, the degree of mixing between classes is exponentially reduced and the distribution becomes broader and consisting of a sequence of three overlapping peaks (one per class).

Let us now study the dependence on the number of classes $N_c$. Fig 7 shows the stationary wealth distribution for different values of $N_c$ and $\beta = 1$ or $\beta = 10$. We notice from the simulations that, the larger the $N_c$ or $\beta$ is, the more generations it takes for the wealth distribution to converge. While in the previous cases, ten generations were enough to go from a Gaussian of average $\mu$ and $\sigma = \mu/10$ to an approximate stationary distribution, for $N_c = 100$ and $\beta = 1$ about 40 generations, and for $N_c = 100$ and $\beta = 10$ about 60 generations.

For $\beta = 1$, as $N_c$ increases, the shape of the distribution changes from an almost flat distribution (with 10 small undulations) for $N_c = 10$, to one with a peak for values below the average

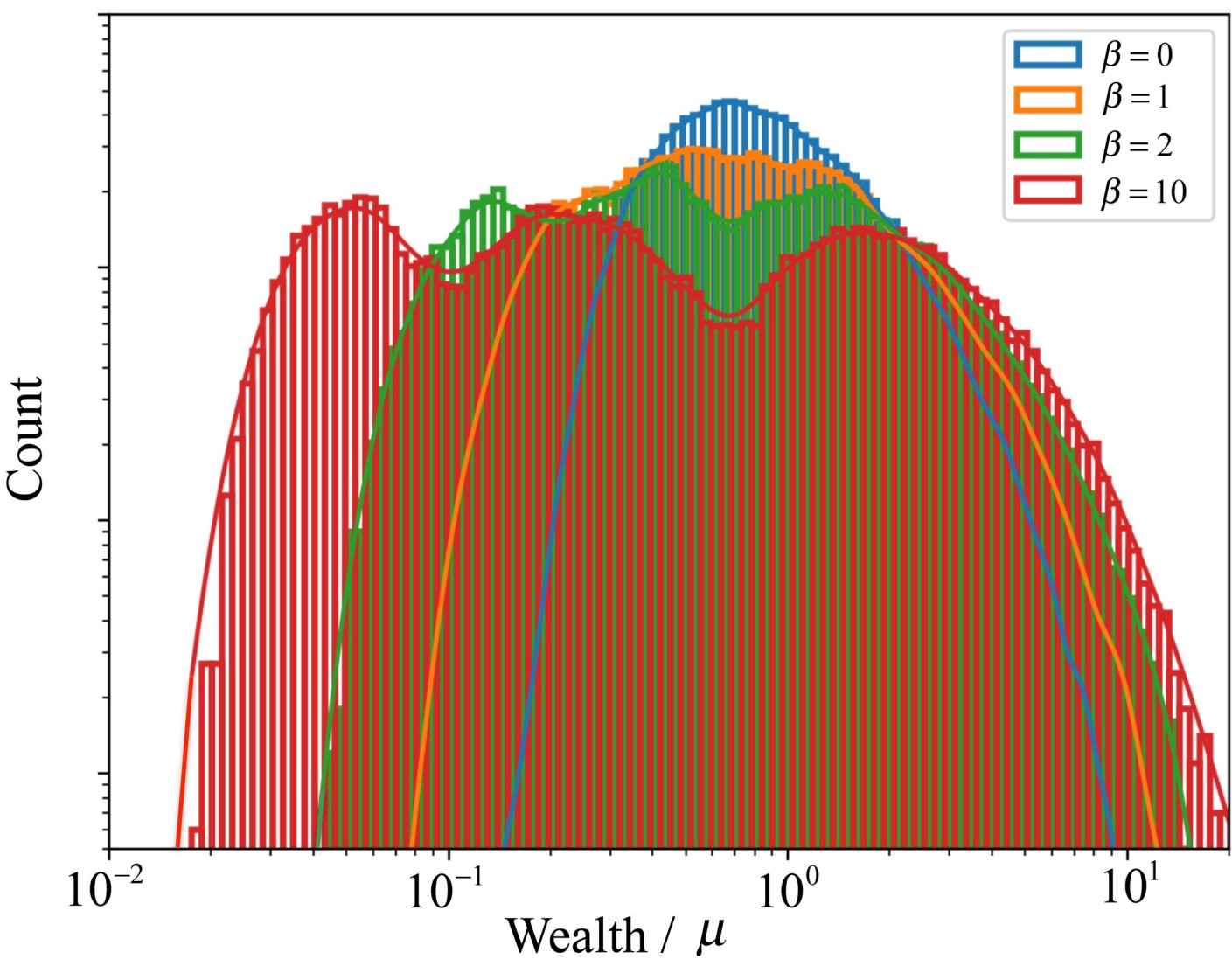

**Fig 6. Stationary wealth distributions for a society with three classes, for $\beta$ = {0, 1, 2, 10} (blue, orange, green, and red) and $N_0 = 2 \times 10^5$.**

wealth $\mu$ and a power-law regime for values of the wealth around $\mu$. The range of the power-law regime widens with $N_c$ and the exponent $\alpha \approx 2/3$ (slope in the log-log plot) seems to be independent of $N_c$. For larger wealth, one observes a second bump that seems to decrease with $N_c$. The Gini coefficient ranges from $G \approx 0.72$ for $N_c = 10$ to $G \approx 0.93$ for $N_c = 100$. The fraction of the wealth in the top 10% individuals is 56% and 93%, respectively.

For $\beta = 10$, the effect of $N_c$ is more pronounced. The Gini coefficient changes from $G \approx 0.87$ for $N_c = 10$ to $G \approx 0.95$ for $N_c = 100$, and the fraction of the wealth in the top 10% individuals is 84% and 95% respectively. As discussed before for $N_c = 3$, the distribution consists of a sequence of $N_c$ peaks. However, the relative height between the peaks is such that, as $N_c$ increases, the overall distribution is consistent with a peak for a value of the wealth below the average and a power law of the same exponent $\alpha \approx 2/3$ as before, independently of $N_c$, and a bump for large values of the wealth. This bump corresponds to the wealthiest class. The height of the bump decreases with $N_c$ and its position moves towards higher values. For $N_c = 100$, the power-law regime extends over three orders of magnitude.

(A)

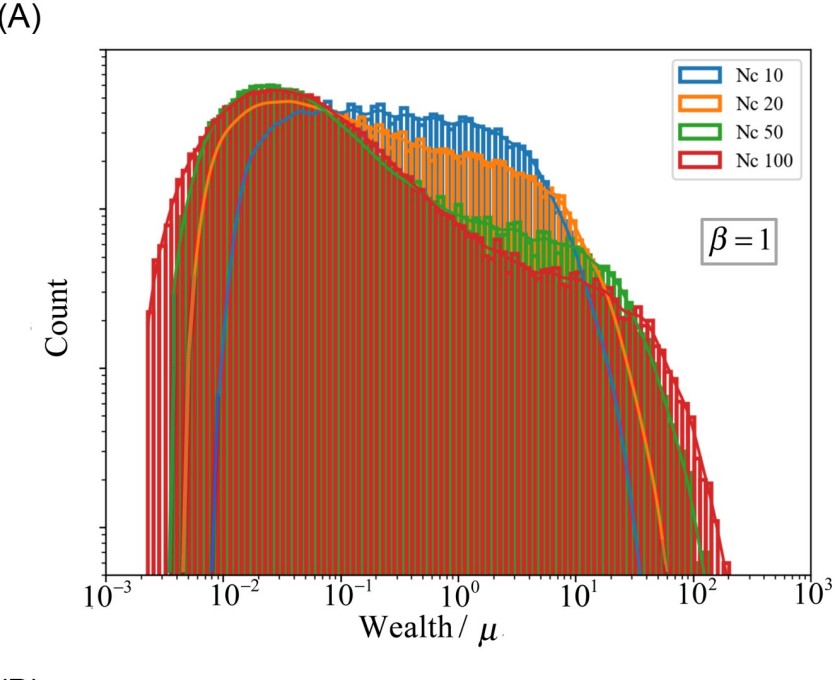

(B)

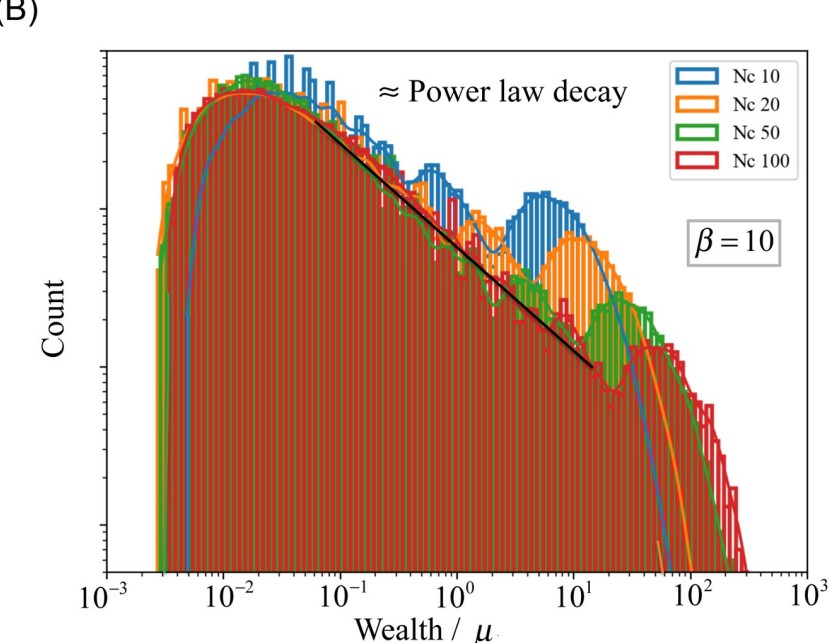

**Fig 7. Stationary wealth distributions for a society with $N_0 = 2 \times 10^5$, $N_c = \{10, 20, 50, 100\}$, and $\beta = 1$ (top) or $\beta = 10$ (bottom).** The (black) solid line corresponds to a power law decay with an exponent $\approx -2/3$.

## IV. Conclusion

The societies we studied here were mainly characterized by a certain offspring probability distribution (Eq 4), accounting for the variable number of children of each family, and a marriage probability distribution (Eq 2) that depends on the society number of different wealth classes $N_c$ and their level of distinction $\beta$.

In a society without classes, where marriages are random, we found (as in [16]) an egalitarian wealth distribution if all families have exactly the same number of children. However, wealth inequality emerges from the moment there is a different offspring distribution, in which families may have different number of children. Both for the more simple example $f^o = \{0, 1/3, 1/3, 1/3\}$ and the more realistic choice $f^o = \{0.1, 0.2, 0.4, 0.2, 0.1\}$, we observed in a few ($\approx 4 - 5$) generations a rapid evolution to a stationary wealth distribution with $G \approx 0.3$. The stationary wealth distribution has, for $w > \mu$, an exponentially decaying (or Boltzmann) law, $f(w) \propto e^{-w/w_c}$, where $w_c$ is a characteristic wealth.

In societies with classes, stationary wealth distributions were also found for the representative offspring distribution $f^o = \{0, 1/3, 1/3, 1/3\}$. The larger the number ($N_c$) and the distinction between classes ($\beta$), the longer it takes to attain the stationary distribution, and the larger is its Gini coefficient. The values we obtained for our model societies reflect the economical empirical known estimates. While Gini coefficients in developed countries typically range between about 0.3 and 0.4 for income, they vary from about 0.5 to 0.9 for wealth [5].

The stationary distributions may, in certain cases, acquire complex and undulated shapes. As the distinction between classes increases, we may observe, in the log-log representation of Fig 5, for a society with only $N_c = 3$ classes, the continuous evolution from a distribution with only one single peak (for $\beta = 0$), to a distribution with $N_c = 3$ overlapping peaks ($\beta = 10$). We also observed in the stationary wealth distributions for other values of $N_c$ the appearance of a number of peaks equal to the society number of classes.

As the number of classes $N_c$ increases, we observe a power-law regime in the stationary wealth distribution, for intermediate values of wealth ($10^{-1} \mu < w < 10^1 \mu$ for $N_c = 100$). This power law exists already for a miscible society (with $\beta = 1$) but it extends over three orders of magnitude for a stricter one ($\beta = 10$). The exponent of the power law remains approximately the same and equal to $\alpha \approx 2/3$, independent of $N_c$ and $\beta$, as soon as these values are large enough.

The exponents reported here differ from the ones observed empirically, that typically range from 1.5 and 3 [2, 3]. Also, note that, in our model, the power-law behavior is observed for intermediate values of wealth and not in the tail for the wealthier. The goal here was to focus on the effect of inheritances and social classes on the wealth distribution. Obviously other factors are also at play and these should be considered in future studies for a more detailed model.

We assumed that every individual can, in principle, marry any other of a different gender (mean field). However, in reality, individuals live in a time-dependent social network and the likelihood of getting married also depends on the effective distance between individuals in such a network. As it is well established, the value of power-law exponents emerging from a non-linear dynamics strongly depend on the spatial dimension and correlations of the underlying topology [20]. How $\alpha$ depends on the underlying topology is a topic of future work.

We also assumed that the sum of the wealth of all individuals remain constant, and that each married couple transmits to its offspring exactly its entire received inheritance. Our model is purely intergenerational. Any wealth evolution during lifecycles is neglected. It would then be interesting to couple our model to lifecycle models, like for example the ones in which the evolution of the wealth of individuals are proportional to their initial wealth, but with a random proportionality constant (the proportional random growth models mentioned in the introduction [8–13]). The global wealth evolution depends in this case on the probability distribution of this proportionality constant. If some small wealth dissipation is included, these models also predict power-law behaviors, with $\alpha > 1$ [7].

In our analysis, we used fixed offspring distributions, that did not change at each new generation. From an historical perspective, we know this is certainly not the case for any society in

our world. Moreover, we mostly considered offspring distributions that preserved, at least approximately, the total population of the society. These offspring distributions led to the stationary wealth distributions we have described in our article. An investigation with variable offspring distributions, allowing the evolution of the population, is straightforward within our model and may also be envisaged in the future.

## Author Contributions

**Conceptualization:** Pedro Patrício, Nuno A. M. Araújo.

**Formal analysis:** Pedro Patrício, Nuno A. M. Araújo.

**Investigation:** Pedro Patrício, Nuno A. M. Araújo.

**Software:** Pedro Patrício.

**Writing – original draft:** Pedro Patrício, Nuno A. M. Araújo.

**Writing – review & editing:** Pedro Patrício, Nuno A. M. Araújo.

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
