## [Decision Letter · Decision Letter 0]

11 Aug 2021

PONE-D-21-21126

Inheritances, social classes, and wealth distribution

PLOS ONE

Dear Dr. Patrício,

Thank you for submitting your manuscript to PLOS ONE. After careful consideration, we feel that it has merit but does not fully meet PLOS ONE’s publication criteria as it currently stands. Therefore, we invite you to submit a revised version of the manuscript that addresses the points raised during the review process.

We look forward to receiving your revised manuscript.

Kind regards,

Haroldo V. Ribeiro

Academic Editor

PLOS ONE

Journal Requirements:

“We acknowledge financial support from the Portuguese Foundation for Science and Technology (FCT) under Contracts no. UIDB/00618/2020 and UIDP/00618/2020.”

“We acknowledge financial support from the Portuguese Foundation for Science and Technology (FCT) under Contracts no. UIDB/00618/2020 and UIDP/00618/2020.”

We note that you have provided funding information within the Acknowledgements Section. Please note that funding information should not appear in the Acknowledgments section or other areas of your manuscript. We will only publish funding information present in the Funding Statement section of the online submission form.

“We acknowledge financial support from the Portuguese Foundation for Science and Technology (FCT) under Contracts no. UIDB/00618/2020 and UIDP/00618/2020.”

Reviewers' comments:

Reviewer's Responses to Questions

**Comments to the Author**

1. Is the manuscript technically sound, and do the data support the conclusions?

Reviewer #1: Yes

Reviewer #2: Yes

2. Has the statistical analysis been performed appropriately and rigorously? 

Reviewer #1: Yes

Reviewer #2: Yes

3. Have the authors made all data underlying the findings in their manuscript fully available?

Reviewer #1: Yes

Reviewer #2: Yes

4. Is the manuscript presented in an intelligible fashion and written in standard English?

Reviewer #1: Yes

Reviewer #2: Yes

5. Review Comments to the Author

Reviewer #1: The paper "Inheritances, social class, and wealth distribution" investigates a model of intergenerational inheritance. The paper is sound and the results are very interesting, in particular, because of the recent discussions on universal income, which certainly would affect wealth distributions.

Despite the complexity of the problem, the authors followed the physics tradition of simplifying the system to one subject to only a few forces. Thus, population growth and wealth growth have been ignored in the model, for the sake of simplicity. The authors also considered that, on average, the ratio of male/female is one, and only same-gender marry is allowed. It would be interesting to see how the same gender marries and differences in the ratio of male/female could affect the intergenerational distribution of wealth.

Despite the simplicity of the model, the authors conclude that even in a society with uniform wealth distribution would become unequal after few generations if the number of children and marriage among people from different classes is less likely than marriage between couples from the same class.

I enjoyed a lot reading this paper and the paper has the merit to be published in PLOS ONE. My only suggestion to the authors is to include a paragraph on the discussion to highlight the limitations of the model and how it could be improved by including population and wealth growth.

Reviewer #2: The paper proposes a model to investigate the impact of inheritance on wealth distribution, considering that wealth is a finite and constant resource equally divided among the offspring over the generations.

The results of the model are interesting and non-trivial.

However, keeping the wealth constant seems to be out of the real situation (new wealth are constantly created), which compromises any possibility of comparing the model's results with the empirical values.

Too bad because the model could achieve something closer to the real world with one or two more considerations.

For example, introducing the fact that an individual born poor can become rich (with a given probability - power law?), or the opposite, and following including new wealth to the system.

It is written briefly in the introduction and conclusion sections that the empirical evidence suggests the exponent alpha varies between 1.5 and 3. However, details are not discussed (for instance, why these values? What is the reason for such dispersion in the values, etc. ).

I suggest creating a new section (e.g. discussion section) exploring empirical evidence and how your model could be compared to them?

That is, in what way to consider constant wealth compromise the quantitative outcome if compared to the empirical evidence?

For instance, the model predicts an exponent ~ 2/3 that is much smaller than the inferior limit of the empirical result.

Other suggestions:

Fig. 7 : I suggest including the value of the power-law exponent in the caption and stress in this caption the robustness of this value. That is an important finding, and it must be clear to the reader.

- I became curious to see two more graphs about the Gini coefficient that could give more insights to the interpretation of the results: i) G as a function of beta, and ii) G as a function of Nc.

You cited these things in the paper, but I think it is more appropriated to present them in a graph/figure.

6. PLOS authors have the option to publish the peer review history of their article (what does this mean?). If published, this will include your full peer review and any attached files.

Reviewer #1: No

Reviewer #2: **Yes: **Fabiano L. Ribeiro

---

## [Author Response · Author response to Decision Letter 0]

27 Sep 2021

Response to Reviewers is included in the resubmission files

---

## [Editor Report · Decision Letter 1]

11 Oct 2021

Inheritances, social classes, and wealth distribution

PONE-D-21-21126R1

Dear Dr. Patrício,

We’re pleased to inform you that your manuscript has been judged scientifically suitable for publication and will be formally accepted for publication once it meets all outstanding technical requirements.

Kind regards,

Haroldo V. Ribeiro

Academic Editor

PLOS ONE

Additional Editor Comments (optional):

I thank the authors for addressing all minor comments from our two reviewers in this amended version of their manuscript. Thus, I consider that there is no need to further review and that the manuscript is ready to be published.

---

## [Editor Report · Acceptance letter]

15 Oct 2021

PONE-D-21-21126R1 

Inheritances, social classes, and wealth distribution 

Dear Dr. Patrício:

I'm pleased to inform you that your manuscript has been deemed suitable for publication in PLOS ONE. Congratulations! Your manuscript is now with our production department. 

Kind regards, 

on behalf of

Dr. Haroldo V. Ribeiro 

Academic Editor

PLOS ONE